# Phase Separation: The Robust Modulator of Innate Antiviral Signaling and SARS-CoV-2 Infection

**DOI:** 10.3390/pathogens12020243

**Published:** 2023-02-03

**Authors:** Yi Zheng, Chengjiang Gao

**Affiliations:** Key Laboratory of Infection and Immunity of Shandong Province, Department of Immunology, School of Basic Medical Sciences, Shandong University, Jinan 250012, China

**Keywords:** phase separation, SARS-CoV-2, RLR, cGAS–STING, N protein, NSP8

## Abstract

SARS-CoV-2 has been a pandemic threat to human health and the worldwide economy, but efficient treatments are still lacking. Type I and III interferons are essential for controlling viral infection, indicating that antiviral innate immune signaling is critical for defense against viral infection. Phase separation, one of the basic molecular processes, governs multiple cellular activities, such as cancer progression, microbial infection, and signaling transduction. Notably, recent studies suggest that phase separation regulates antiviral signaling such as the RLR and cGAS–STING pathways. Moreover, proper phase separation of viral proteins is essential for viral replication and pathogenesis. These observations indicate that phase separation is a critical checkpoint for virus and host interaction. In this study, we summarize the recent advances concerning the regulation of antiviral innate immune signaling and SARS-CoV-2 infection by phase separation. Our review highlights the emerging notion that phase separation is the robust modulator of innate antiviral signaling and viral infection.

## 1. Introduction

Sensing of pathogen-associated molecular patterns (PAMPs) by the corresponding pathogen recognition receptors (PRRs) is the first step in innate immunity signaling [1]. Three kinds of PRRs, including Toll-like receptors (TLRs), RIG-I-like receptors (RLRs), and cytosolic DNA sensors (CDSs) such as cyclic GMP-AMP synthase (cGAS), are majorly responsible for the recognition of the viral PAMPs [1]. Different TLRs can recognize DNA and RNA, while RLRs and cGAS are responsible for the detection of RNA and DNA, respectively. The perception of viral nucleic acids by PRRs initiates the recruitment and activation of the adaptor protein such as Toll/interleukin-1 receptor domain-containing adaptor protein inducing interferon-beta (TRIF), myeloid differentiation factor-88 (MyD88), mitochondrial antiviral signaling protein (MAVS), and stimulator of interferon genes protein (STING) [2]. TRIF and MyD88 are the downstream adaptor proteins of TLR receptors, while MAVS and STING are the adaptor proteins for RLRs and cGAS, respectively. The aggregation and activation of adaptor proteins act as a scaffold to recruit and activate the kinases including inhibitor of nuclear factor kappa-B [IκB] kinase (IKK) family and TANK-binding kinase-1 (TBK1). Furthermore, the IKK family and TBK1 are responsible for the activation of transcription factors nuclear factor kappa-light-chain-enhancer of activated B cells (NF-κB) and interferon-regulatory factor 3 (IRF3), respectively [3]. The activation of transcription factors leads to their nuclear translocation to stimulate the expression of the proinflammatory cytokines and type I interferons (IFNs) (Figure 1).

SARS-CoV-2, the causative agent of COVID-19, has led to a worldwide pandemic since 2019 [4]. As a highly pathogenic human coronavirus, SARS-CoV-2 possesses multiple strategies to antagonize the antiviral host responses to establish infection and spread. A typical characteristic of COVID-19 patients is that antiviral immunity is inhibited, but proinflammatory responses are stimulated [5]. Until now, extensive studies have suggested that structural, nonstructural, and accessory proteins of SARS-CoV-2 are involved in dampening the type I interferon response [6,7,8,9,10]. Moreover, in vitro and in vivo studies demonstrate that the treatment with IFNs at appropriate time points and doses could facilitate the recovery from the viral infection [11]. These studies indicate that the arms race between innate antiviral immune signaling and SARS-CoV-2 is a critical event during viral infection, likely determining the outcome.

Phase separation, either liquid–liquid phase separation (LLPS) or liquid–gel phase separation, is considered the driving force for biomolecular condensates. Biomolecular condensates are critical for the formation of membrane-less organelles (MLOs) such as nucleoli, Cajal bodies, promyelocytic leukemia nuclear bodies (PML-NBs), and stress granules [12,13,14,15]. Many of the proteins within the above MLOs exhibit features of nucleic acid-binding regulatory proteins and are composed of proteins and RNAs. Therefore, MLOs represent platforms for the regulation of gene expression in both the nucleus (such as nucleoli, Cajal bodies, and PML-NBs) and the cytoplasm (such as stress granules) [13,16,17]. MLO formation contributes to the intracellular compartmentalization of specific biological functions. A recent study also suggests that phase separation can happen with the membranes [18]. Phase separation is driven by the multiple weak interactions, including intrinsically disordered regions (IDRs), the RNA/DNA-binding domains, and the interaction between proteins and nucleic acids RNA/DNA [12,13]. IDRs lack a defined 3D structure but often contain repeated sequence elements that provide the basis for multivalent weakly adhesive intermolecular interactions [12]. The intermolecular or intramolecular protein–protein and protein–RNA interactions contribute to the multivalency that promotes MLO formation [12]. Biomolecular condensates are involved in multiple cellular activities, such as cancer progression, gene expression, and signaling transduction [15]. Viral proteins also form condensates which are required for the replication of various viruses, such as the measles virus, human respiratory syncytial virus (RSV), and SARS-CoV-2 [19,20,21]. These studies found that small-molecule drugs blocked viral replication by affecting the LLPS of viral proteins. The antiviral drugs enhancing the liquid phase to solid phase conversion of viral proteins or disrupting the LLPS of viral proteins can significantly affect viral replication [21,22]. Phase separation also participates in regulating the activities of essential signaling molecules to fine-tune the antiviral signaling cascade [23]. These phenomena suggest that phase separation likely plays multiple roles in regulating the viral–host interaction.

In this review, we summarize the recent progress concerning the regulation of antiviral innate immune signaling and SARS-CoV-2 infection by phase separation. We especially emphasize the phase separation of the essential signaling molecules such as cGAS, STING, and IRF3 as well as the nucleocapsid protein (N) and non-structural 8 (NSP8) of SARS-CoV-2. The phase separation of viral proteins involved in modulating antiviral response is also discussed. Finally, we list the possible small molecules targeting the viral proteins, which are the potential drugs for the treatment of viral infections.

## 2. Phase Separation of cGAS and STING

cGAS, with 522 amino acids and 62 kDa, is the major receptor for dsDNA and microbial DNA [24]. cGAS senses abnormal cytosolic dsDNA derived from pathogens or from nuclear or mitochondrial damage. Exogenous DNA leads to the cytoplasmic foci formation containing cGAS and DNA [25]. cGAS is composed of the N-terminal domain (NTD), middle NTase core, and C-terminal Mab21 domain. NTD is positively charged and disordered (residues 1–160), while the core part of enzymatic activity (NTase core, residues 161–330), which catalyzes the synthesis of cyclic GMP-AMP (cGAMP), partially overlaps with the Mab21 domain (residues 213–513) (Figure 2A). Residues from 389 to 405 represent a zinc-ribbon domain for binding with zinc, which stabilizes the DNA binding and cGAS dimers [26,27]. Recently, DNA binding to cGAS is shown to induce the formation of liquid-like condensates [25]. For the cGAS-dsDNA liquid droplets, the essential multivalent elements to drive its LLPS include the positively charged N-terminal domain of cGAS, long DNA strands, and free zinc ions. As aforementioned, zinc stabilizes the interaction between DNA and cGAS to enhance the multivalency to promote the LLPS of cGAS [25]. A recent study also identifies a novel DNA-binding interface of cGAS, which facilitates the LLPS of cGAS [28]. Functionally, the LLPS of cGAS promotes its enzymatic activity by protecting DNA from degradation by the exonuclease TREX1 [29] (Figure 1). In addition to DNA, cGAS also forms liquid-like condensates with dsRNA, although dsRNA does not activate cGAS to produce cGAMP [25]. A recent report demonstrates that a high concentration of dsRNA interferes with cGAS binding to DNA to inhibit cGAS activity, whereas dsRNA at low concentration facilitates phase separation and the production of cGAMP [30].

The condensate formation of cGAS is regulated by multiple cellular and viral proteins (Figure 1). GTPase activating protein (SH3 domain) binding protein 1 (G3BP1), the hub protein for stress granule assembly, is shown to promote the DNA binding and oligomerization of cGAS [31]. A recent report demonstrates that G3BP1 promotes the gel-like condensation of cGAS before DNA stimulation, which makes the cGAS polymerization ready for the DNA-induced LLPS [32]. Ubiquitin-Specific-Processing Protease 15 (USP15), a protease involved in protein deubiquitination, promotes the cGAS–STING signaling through two independent mechanisms. One way is that USP15 catalyzes the deubiquitination of cGAS dependent on its enzymatic activity. On the other hand, USP15, through its IDR region, can promote the condensate formation of cGAS [33]. In contrast to G3BP1 and USP15, poly(RC)-binding protein 2 (PCBP2) impedes the LLPS of cGAS and negatively regulates the cGAS–STING signaling. PCBP2 interacts with cGAS, decreases cGAS enzymatic activity, and impairs the cGAS–STING signaling by antagonizing cGAS condensation [34]. Since cGAS provides an essential platform for viral DNA recognition and subsequent viral clearance, viruses evolve to antagonize the activity of cGAS. Herpes virus encodes tegument protein ORF52/VP22, a DNA-binding protein, to compete with cGAS for DNA-binding. This competition disrupts the multivalent interaction between cGAS and dsDNA, which further impairs the LLPS of cGAS [35]. Different from PCBP2, ORF52 inhibits the LLPS of cGAS through binding with dsDNA instead of interacting with cGAS. These studies suggest that proteins can modulate the LLPS of cGAS by interacting with either cGAS or dsDNA.

The STING protein, with 379 amino acids and 42 kDa, is the critical adaptor protein in cGAS–STING signaling. STING resides on the ER membrane in the resting state. The second messenger cGAMP, which is produced by cGAS upon dsDNA stimulation, binds to STING, stimulating its ER-Golgi translocation, aggregation, and TBK1 activation to induce the downstream type I interferon signaling [36,37,38] (Figure 1). STING is composed of a cytosolic N-terminal tail, N-terminal four transmembrane (TM) domains, dimerization domain (DD), ligand-binding domain (LBD), and C-terminal tail (CTT). The IDR region (residues 309–343) is located within the LBD domain [18,36,39] (Figure 2B). A recent study suggests that STING forms liquid-like droplets in vitro without cGAMP [18]. The addition of cGAMP induces the gel-like phase transition of STING. STING phase separation is dependent on the C-terminal IDR and dimerization domain, which provides the multivalent interaction critical for phase separation. Within IDR, E^336^/E^337^ are the two essential residues for phase separation. Moreover, the transmembrane domains of STING are also essential for its droplet formation since STING forms jigsaw puzzle-like condensates within the ER membrane [18]. Functionally, the phase separation of STING occurs in the late time of DNA viral infection and negatively regulates innate immune signaling. Mechanistically, unphosphorylated TBK1 is recruited into the droplets of STING as the client protein. The complex containing TBK1 and STING insulates and prevents the IRF3 from interacting with TBK1 [18]. Another study demonstrates that a polyvalent STING agonist PC7A can induce the STING–PC7A condensate [40]. Distinct from cGAMP-induced condensate, this complex of STING–PC7A includes the phosphorylated TBK1, suggesting this condensate promotes downstream signaling [40]. Consistently, PC7A induces the prolonged production of proinflammatory cytokines by binding to a non-competitive STING surface site different from the cGAMP binding pocket. A combination of PC7A and cGAMP leads to a better anti-tumor activity dependent on STING. These two studies suggest that STING likely forms two types of condensate that play either a positive or negative role in mediating type I interferon signaling [18,40,41].

## 3. Phase Separation in Regulating the RLR Signaling

RIG-I and MDA5 are the major RLR receptors (RLRs) for recognizing viral RNAs. The interaction of viral RNA with RLRs leads to its oligomerization and interaction with the downstream adaptor protein MAVS. The interaction of caspase activation and recruitment domains (CARDs) of RLRs with CARD of MAVS further results in MAVS aggregation, fibril formation, and downstream signaling activation [42,43,44] (Figure 1). With the stimulation of viral RNA or mimic, the host cell forms the stress granules in which RLR receptors and downstream signaling molecules are located [45]. As aforementioned, the LLPS of G3BP1 is the driving force of stress granules [46]. Two recent studies suggest that G3BP1 also associates with RIG-I to facilitate its binding to dsRNA and downstream signaling pathways [47,48] (Figure 1). Whether RIG-I is the client protein of G3BP1 droplets in the context of viral RNA stimulation warrants further investigation. A recent study also suggests that the critical adaptor protein MAVS also undergoes LLPS to form condensates, which are counteracted by the LLPS of the N protein of SARS-CoV-2 [22] (Figure 3). Functionally, the LLPS of MAVS is critical for its prion-like formation, but which domain of MAVS is critical for its LLPS is currently unknown.

## 4. Phase Separation of Transcriptional Factors IRF3 and IRF7

IRF3, with 427 aa and 50 kDa, is the critical transcriptional factor in RLR and cGAS–STING signaling. It is phosphorylated by the upstream kinase TBK1/IKKε in the cytoplasm, which promotes its dimerization and translocation. After translocation into the nucleus, dimerized IRF3 binds to the regulatory elements of IFN gene promoters to stimulate the expression of type I interferons (Figure 1). IRF3 consists of four modules from N to C terminus: DNA-binding domain (DBD) for binding with the regulatory elements of IFN gene promoters, a proline-rich linker which is an IDR region according to software prediction, IRF-associated domain (IAD) for dimerization, and signal response domain (SRD) containing serine residues 386 and 396 for activation (Figure 2C). The SRD domain is inhibitory in the absence of phosphorylation [49].

In vitro assays indicate that IRF3 forms liquid-like droplets [50]. Domain mapping demonstrates that DBD is required for its LLPS, while IAD and IDR contribute to LLPS to a less extent. The SRD domain plays a negative role in LLPS, and the deletion of SRD significantly enhances its LLPS. Similar to results in vitro, cellular study suggests that viral infection leads to condensate formation of IRF3 in the nucleus. Interferon-stimulated response element (ISRE) DNA and IRF7 promote the LLPS of IRF3 [50]. Moreover, SIRT1 catalyzes the deacetylation of specific lysine residues in the DBD of IRF3 and IRF7, which is necessary for their LLPS and transactivation of type I interferons [50] (Figure 1 and Figure 2C).

Neurofibromatosis protein 2 (NF2) is a classic tumor suppressor. Another study indicates that missense mutations of the tumor suppressor neurofibromin 2 (NF2m) form condensates in the cytoplasm upon activation of the innate immune signaling [51]. Interestingly, this condensate contains TBK1, IRF3, and PP2A. The phosphatase PP2A catalyzes the dephosphorylation and inactivates TBK1. NF2 mutation recruits IRF3 5D, the constitutive form of IRF3, into the liquid-like droplets. This condensate formation prevents IRF3 from translocating into the nucleus. Therefore, the NF2 mutation droplets robustly suppress the activation of innate immune signaling [51].

## 5. Phase Separation of Nucleocapsid (N) Protein of SARS-CoV-2

Coronavirus virions encode four structural proteins that form the virion: nucleocapsid (N), envelope (E), membrane (M), and spike (S). The N protein of the coronavirus is the most abundant protein in infected cells and is strongly immunogenic [52]. The N protein of SARS-CoV-2 is a multifunctional protein [53,54] involved in viral genome packaging/virion assembly [55,56], viral replication and transcription [57,58,59], inflammation [60,61], and type I interferon response [10,62,63] (Figure 3). The N protein is composed of 419 amino acids, which is a 49 kDa RNA-binding protein [54]. The N protein of SARS-CoV-2 has a modular organization, including intrinsically disordered regions (IDRs) and conserved structural regions according to the sequence characteristics [53,64,65]. The five modules are divided into the N arm (N_IDR_), the N-terminal domain (NTD), the linker region (Linker_IDR_), the C-terminal domain (CTD), and the C tail (C_IDR_) (Figure 2D). The Linker_IDR_ is also known as the serine-arginine (SR)-rich linker region, and the phosphorylation of this region affects the viral transcription and replication, which will be discussed later. NTD and CTD domains are both responsible for binding with viral or cellular RNA [65]. A recent study demonstrates that a structured sequence such as transcription regulatory sequence (TRS) is recognized by the NTD for subgenomic RNA generation or sgRNA encapsidation, whereas the CTD domain recognizes the dsRNA independent of its sequence [66]. This suggests that the recognition of RNA by NTD or CTD may be related to the multiple functions of the N protein. The CTD domain is also a dimerization domain and forms a highly stable dimer, and the N protein or CTD domain of the N protein is purified as the dimer or oligomer state contributed by CTD [64,67]. Due to the disordered IDR region of the N protein, the structure of a full-length N protein remains to be elucidated [67]. However, the structures of NTD and CTD domains have been solved by the crystallization-based method [67,68]. The NTD domain has a right-handed fist shape consisting of an antiparallel β-sheet core subdomain and a protruding β-hairpin region. The CTD domain is present as a tightly intertwined homodimer and displays an overall rectangular slab shape [67]. In a recent study, the structure of a full-length N protein with RNA or without RNA has been shown using electron microscopy and molecular dynamics simulations. They demonstrate that the N protein forms structurally dynamic dimers with extended conformations in the absence of RNA, while the presence of RNA makes the N protein assume a more compact conformation where the NTD and CTD are packed together reminiscent of viral ribonucleoproteins (vRNP) particles [69].

Previous studies suggest that IDR is crucial for driving the robust phase separation of proteins [12]. Given the prevalence of the IDR regions in the N protein, the N protein is a competent protein for phase separation [53,70]. A mixture of a purified N protein with cellular or viral genomic RNA leads to condensate formation. Even though there is little consensus about which domain of the N protein is required for its phase separation, LinkerIDR and CTD domains are generally considered essential for phase separation [56,57,63,65,71,72,73] (Figure 2D). The RNA sequence and structure are important for driving the phase separation of the N protein. Studies have suggested that the 5′ end or 3′ end of the SARS-CoV-2 genome sequences with N protein promote its phase separation, while the frameshift region dissolves the N protein condensates [74]. As aforementioned, the structured sequence such as TRS or putative packaging signal is recognized by the NTD of the N protein for subgenomic RNA generation or sgRNA encapsidation [66]. These studies suggest that the multiple functions of the N protein may be mediated by the interaction between the different domains of the N protein with distinct sequences and structures of RNA [66,74]. The phase separation of the N protein is directly or indirectly related to the aforementioned functions of the N protein [53]. We will focus on its involvement in the viral genome packaging/virion assembly as well as viral replication and transcription in this section. The association between phase separation of the N protein with inflammation and type I interferon response will be discussed in the later section.

After infection, the RNA genome of the SARS-CoV-2 virus is released into the cytosol, translated, and cleaved into individual nonstructural proteins (NSPs) by viral proteases [75]. The NSPs together form the replication transcription complex (RTC), and viral genomic replication and subgenomic mRNA transcription are initiated in double-membrane vesicles (DMVs) [53,76]. In infected cells, the N protein is localized to the vicinity of RTC and is a cofactor of RTC [77]. NSP3 is an essential RTC component, and a recent study suggests that NSP3 forms the pores of RTC by electron tomography and facilitates the release of the viral genome from DMV [78]. The N protein is shown to interact with NSP3 through the interaction between the Linker_IDR_ of the N protein and the ubiquitin-like (Ubl) domain of NSP3 [58]. The N protein located in the vicinity of NSP3 likely facilitates the encapsidation of exited naked RNA and the protection of viral RNA from detection by the host defense system [58]. More interestingly, another study suggests that the NSP3 Ubl domain is recruited into the N-RNA droplets in vitro [70]. Therefore, the N protein likely plays a role in viral replication and transcription through concentrating NSP3 in RTC. The viral replication and transcription are mediated by the holo–RdRp complex, which is composed of the core enzyme NSP12 as well as the cofactor NSP7 and NSP8 [75]. In another two studies, N protein–RNA condensates can recruit the NSP12, NSP7, and NSP8 [57,79], suggesting another regulatory role of N protein phase separation in viral replication and transcription. Post-translational modification (PTM), especially phosphorylation, plays an important role in the modulation of N protein involvement in viral replication and transcription [80,81,82]. A major site of phosphorylation is the SR-rich linker region within the Linker_IDR_, which is catalyzed by the host kinases shortly post-infection [83]. Studies suggest that the phosphorylation of serine residues in an SR-rich motif enhances the dynamic kinetics of complex including N protein and RNA, with unmodified N protein forming gel-like condensate with viral RNA [56,70]. A recent study also indicates that the phosphorylation of the N protein affects its interaction with viral genomic RNA [84]. Moreover, inhibition of the phosphorylation of N strongly affects viral replication [82], while the N protein mutations augment replication and pathogenesis by enhancing the phosphorylation level of the N protein [80,81]. These studies suggest that phase separation of phosphorylated N protein is dynamically involved in regulating viral replication and transcription likely through concentrating various RTC components such as NSP3, NSP7, NSP8, and NSP12 (Figure 3, middle left panel).

During the replication of the viral RNA, the SARS-CoV-2 N protein accumulates in cytoplasmic complexes along the membrane of the Golgi and other vesicles to initiate the process of virion assembly in cooperation with the membrane (M) protein [85]. The DMVs, Golgi membrane, and vesicular structures appear to be clustered near one another, presumably to coordinate RNA synthesis and packaging, which is followed by virus budding, virion assembly, and virion release [86,87]. After virion release, the N protein binds to the 30 kb RNA genome to form vRNPs, heterogeneous structures with N protein–RNA complexes tightly packed in a cylindrical or “bucket-like arrangement” when visualized using high-resolution imaging [71,88,89]. During the virion assembly, the interaction between N and M proteins is critical [90]. A recent study resolves the cryo-electron microscopy structure of the SARS-CoV-2 M protein in long isoform and short isoform conformations, with both isoforms showing a C2 symmetric dimeric structure [55]. In the same study, the authors also demonstrate that the intravirion domain of the M protein is responsible for its interaction with the N protein, and RNA synergistically enhances their interaction [55]. Another study indicates that the M protein of SARS-CoV-2 stimulates the phase transition of N even without RNA [56]. Different from the LLPS of N protein with non-specific 17mer RNA, theM protein induces the gel-like formation of the N + M complex. Interestingly, the N protein, M protein, and RNA form an exclusive complex with N + M or N + RNA, suggesting a stepwise virion assembly [56]. These studies suggest that N–RNA condensates likely synergistically enhance the interaction between N protein and M protein to facilitate the efficient packaging of viral RNA [55,56,91] (Figure 3, left panel). Within the virion, the N protein forms viral RNPs with viral RNA. It has been indicated that unmodified N protein, when combined in vitro with short fragments of the viral genome, forms partially ordered gel-like condensates and discrete 15 nm particles similar to the vRNP structures observed within virions [70,84,89]. The phosphorylation level of the N protein seems to be a critical switch of phase separation status of the N protein to determine its functions, with unmodified N protein promoting the virion assembly (hypophosphorylated, high viscosity) and phosphorylated N protein recruiting the RTC factors for efficient viral replication (hyperphosphorylated, low viscosity) [56,70,84].

## 6. Phase Separation of Non-Structural Protein 8 (NSP8) of SARS-CoV-2

As aforementioned, NSP8 and NSP7 form the RdRp complex with the core enzyme NSP12 [75]. NSP8 protein, with 198 amino acids and 21.88 kDa, confers the processivity to NSP12 together with NSP7 [75] (Figure 2E). The NSP8 protein is composed of an N-terminal extension domain and a C-terminal head domain [92]. The cryo-electron microscopy structure of the SARS-CoV-2 RdRp demonstrates that two copies of the C-terminal head domain of NSP8 (residues 99–198) are responsible for interaction with NSP7 and NSP12 to form the RdRp complex, while the N-terminal extension domain (residues 1–98) functions as the slide pole of the RNA template-product duplex with its positively charged residues [92,93,94] (Figure 2E). Besides the RdRp complex, other NSPs such as NSP13 helicase are also recruited to RTCs. The N-terminal region of NSP8 is shown to interact with NSP13 [95] (Figure 2E). A recent study suggests that NSP8 possesses the property of phase separation [96]. A Predictor of Natural Disordered Regions (PONDR) analysis of the NSP8 peptide sequences demonstrated that the N-terminal 76 residues of the NSP8 protein are likely an IDR region [96]. Purified NSP8 protein was identified as tetramer and dimer through gel filtration analysis [96]. The in vitro droplet assay indicates that the dimer form of NSP8 undergoes LLPS even without RNA, while the tetramer of NSP8 transit from a solid-like state to a liquid-like droplet with an appropriate amount and length of viral RNA. The IDR region is critical for the LLPS of NSP8, and deletion of the N-terminus of NSP8 significantly impairs the droplet formation of NSP8 [96]. The LLPS of NSP8 is promoted by a low concentration of sodium chloride, and a high salt concentration abolishes the LLPS of dimer and tetramer. Besides showing the LLPS property in vitro, the authors find that NSP8 also undergoes LLPS in cellulo. Given the structural study demonstrating the interaction between NSP8 and NSP7 or NSP12 [92], it will be interesting to examine whether NSP8 droplets recruit NSP7, NSP12, and NSP13 as the client proteins. The LLPS property of NSP8 likely increases the concentration of the RdRp complex and helicase NSP13 for efficient transcription and replication. Moreover, the N protein is recruited to the RTC complex by NSP3 [58], the transmembrane protein forming a pore on the DMV [78]. Previous studies have suggested that coronavirus replication occurs near the membrane [75]. Therefore, whether NSP3 recruits the N protein and RdRp complex to the RTC machinery through LLPS warrants further investigation.

## 7. The Arms Race between SARS-CoV-2 and Antiviral Innate Immunity Regulated by Phase Separation

A characteristic feature of COVID-19, the disease caused by SARS-CoV-2 infection, is the dysregulated immune response with impaired type I and III interferons (IFN) expression and an overwhelming inflammatory cytokine storm [5]. Current studies suggest that the phase separation of the N protein is involved in decreasing interferon production and enhancing inflammation.

A proteomic study indicates that the N protein interacts with stress granule proteins [97]. SG is a critical MLO for innate immune response and is formed through the LLPS of its core protein G3BP1 and other client proteins [46]. The residues 1–25 of the N protein interact with the NTF2-like domain of G3BP1 [98]. Interestingly, an in vitro study demonstrates that N protein partitions into phase-separated forms of full-length human hnRNPs (TDP-43, FUS, hnRNPA2) and their low-complexity domains (LCs), which are the SG components [65]. The N protein is observed to phase separately with G3BP1 to promote the disassembly of the stress granule [72], consistent with another two studies [9,62]. The arginine methylation of the N protein by protein arginine methyltransferase 1 (PRMT1) is critical for inhibiting stress granule formation [99]. Given that G3BP1 or a stress granule is crucial for driving innate immunity, the phase separation of N with G3BP1 likely sequesters G3BP1 from other SG components and impairs innate immunity [9]. Another study indicates that the N protein affects the RLR signaling by targeting the critical adaptor protein MAVS [22]. Mechanistically, the phase separation of the N protein strongly inhibits the LLPS of MAVS, affecting the prion-like aggregation of MAVS and subsequent downstream signaling. The C-terminal dimerization domain is critical for the phase separation of the N protein and its antagonism of MAVS LLPS. In addition, acetylation and phosphorylation of the N protein contribute to counteracting the type I interferon response [22,63].

The severity and mortality of COVID-19 are closely associated with virus-induced over-activated inflammatory responses and cytokine storms. However, the exact molecular mechanism governing the inflammation with viral infection is not understood. Recently, the N protein is shown to enhance the activation of NF-κB signaling in the context of viral infection [60]. Mechanistically, the N protein undergoes LLPS with RNA and forms condensates. This condensate of the N protein recruits TAK1 and IKK complex, the essential kinases of NF–κB signaling, as the clients’ proteins to facilitate NF-κB activation [60]. Another study implicates that the N protein also enhances the NLRP3 inflammasome activation, but whether phase separation of the N protein is involved in this process is unknown [61].

## 8. Small Molecules Targeting the Phase Separation of Innate Immune Signaling and SARS-CoV-2

Phase separation of the innate immune signaling molecules plays positive and negative roles in the signaling cascade. Therefore, targeting the phase separation of innate immune signaling molecules needs to be specifically designed. Even though one study suggests that STING condensate formation plays a negative role in interferon signaling [18], another shows that PC7A, a polyvalent STING agonist, facilitates the formation of STING condensates and prolongs the activation of innate immunity pathways [40]. Thus, PC7A leads to synergistic therapeutic outcomes in vivo when combined with the STING ligand cGAMP [41]. Deacetylation of IRF3 by SIRT1 is shown to be required for its LLPS to induce the expression of type I interferons. Furthermore, the study shows that SIRT1 agonists, such as resveratrol (SRT501) and a chemically synthetic compound SRT2183, rescued SIRT1 activity in aged mice, restored IFN signaling, and thus antagonized viral replication. Therefore, SIRT1 agonists may be a treatment method for curing viral infection in the aged population [50]. Some autoimmune disease is driven by IRF3-mediated expression of interferon and downstream ISGs [100]. Thus, SIRT1 inhibitors such as EX527 may be a potential strategy for treating autoimmune diseases [50].

The phase separation of the N protein is required for viral genome packaging, RTC formation, and counteracting host cell immune response. Multiple small molecules or peptides to disturb or facilitate the N condensate have been discovered [22,63,79]. GCG potently impairs the phase separation of the N protein and suppresses SARS-CoV-2 replication [63]. A peptide targeting the dimerization domain (CTD) disrupts the condensates of the N protein to inhibit SARS-CoV-2 replication and rescue innate antiviral immunity in vitro and in vivo [22]. Given the high toxicity of 1,6-hexanediol, developing a low-toxicity molecule targeting N protein phase separation is a strategy to attenuate the inflammatory storm caused by COVID-19 [60]. Besides attenuating the phase separation of the N protein, enhancing the phase separation of the N protein likely is another strategy to restrict viral replication. CVL218 or PJ34 enhances the liquid-like property of the N protein condensates or N-NSP12 protein condensates. Interestingly, the authors find that these chemicals promote the suppression of viral infection in combination with remdesivir, which targets NSP12 [79]. This is likely because that CVL218 or PJ34 attenuates the local density of the condensates and thus promotes the entrance of other antiviral drugs into their targets.

## 9. Conclusions and Perspectives

Even though phase separation is a critical modulator of gene expression, the function and molecular mechanism of phase separation in regulating innate immune signaling and emerging viral infection are still in their infant stage. Recently, a series of essential signaling molecules with phase separation properties was identified. More importantly, their LLPS or phase separation is critical for modulating the signaling transduction. The emergence of SARS-CoV-2 is a pandemic threat to the worldwide population. Targeting the viral RdRp or N protein is a promising strategy for combating the viral infection based on the low mutation of these proteins. Since N and NSP8 possess the phase separation property, small molecules targeting the condensates of N and NSP8 proteins are likely effective in inhibiting the viral infection. A couple of molecules based on the phase separation of N have been developed and found to be effective in vitro [22,63,79]. How to modify these lead compounds to inhibit viral replication in vivo should be investigated in the future.

Proper innate immunity is beneficial for combating viral infection and cancer immunity, but overactivation of innate immunity may lead to autoimmune disease and even neuronal diseases. Phase separation targeting innate immunity may have a potential for treating the disorders listed above. As aforementioned, the NF2m leading to the condensate formation abrogates the TBK1–IRF3 signaling, which contributes to cancer immune evasion. On the other hand, autoimmune diseases related to self-DNA recognition rely on the cGAS–STING-IRF3 pathway to induce type I interferon production [101,102]. Therefore, it is conceivable to apply the molecules targeting the phase separation to treat autoimmune diseases. Phase separation may provide a new pathway for designing drugs to combat a variety of diseases.

## Figures and Tables

**Figure 1 pathogens-12-00243-f001:**
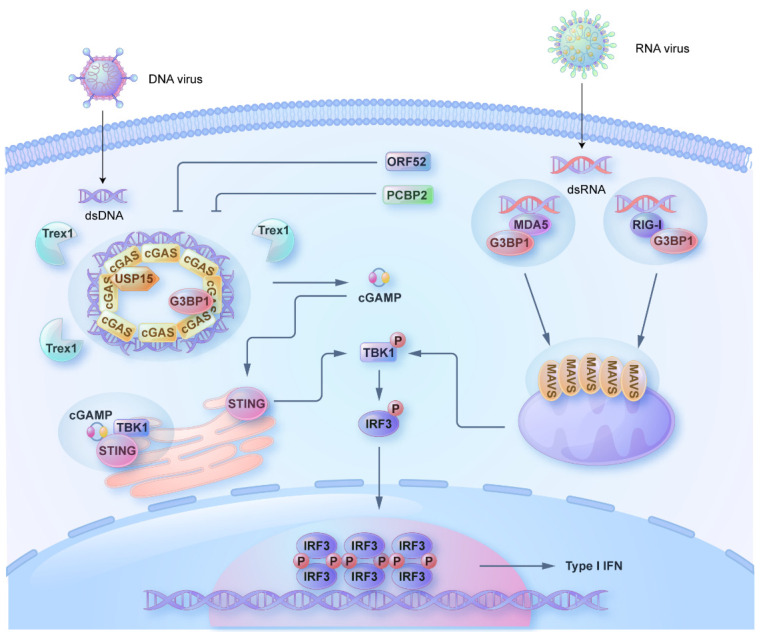
Phase separation in the regulation of antiviral innate immune signaling. The liquid–liquid phase separation of cGAS protects DNA from the degradation of Trex1 to enhance the cGAMP production, which is promoted by G3BP1 and USP15 and antagonized by ORF52 and PCBP2. cGAMP interaction with STING leads to its trafficking from ER to Golgi for downstream TBK1 activation, but part of cGAMP also induces the gel-like droplets of STING, which includes unphosphorylated TBK1 on the ER membranes. Stress granules driven by the LLPS of G3BP1 include RIG-I and MDA5. The activation of RIG-I and MDA5 further stimulates the aggregation of MAVS on the mitochondria in liquid-like droplets. cGAS–STING signaling and RLR-MAVS signaling are converged on the kinase TBK1 and transcriptional factor IRF3. Phosphorylated and dimerized IRF3 translocates into the nucleus and forms condensates with the ISRE DNA element to drive the type I interferon response.

**Figure 2 pathogens-12-00243-f002:**
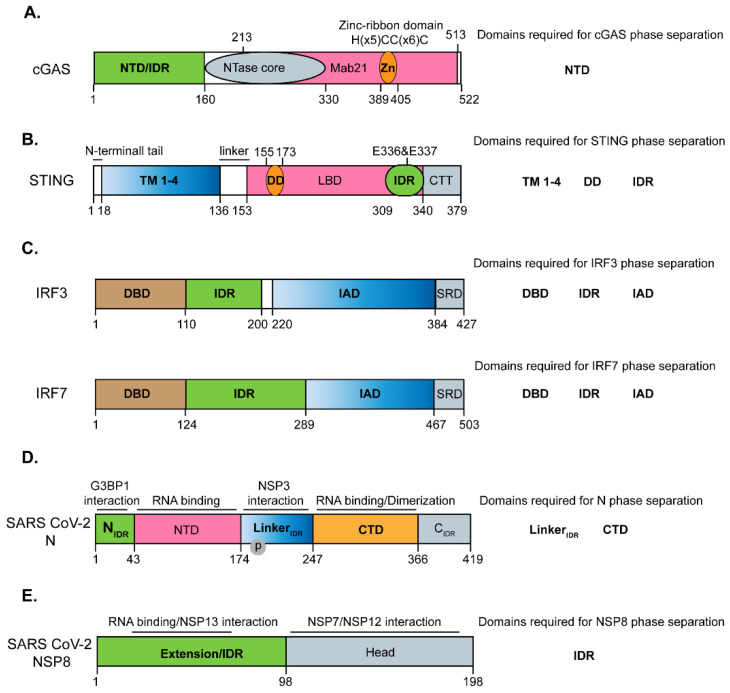
The domain structure of the phase-separated proteins discussed in this study and the domains required for their phase separation. (**A**) The domain structure of cGAS. cGAS is composed of the N-terminal IDR, middle NTase core, and C-terminal Mab21 domain. The N-terminal IDR region is required for its phase separation. The NTase core (residues 161–330) is the core enzymatic part for the synthesis of cGAMP. Residues from 389 to 405 represent a zinc-ribbon domain for binding with zinc, which stabilizes the DNA-binding and cGAS dimers. (**B**) The domain structure of STING. STING is composed of a cytosolic N-terminal tail, N-terminal four transmembrane (TM) domains, dimerization domain (DD), ligand-binding domain (LBD), and C-terminal tail (CTT). The IDR region (residues 309–343) is located within the LBD domain. TM, DD, and IDR are critical for the phase separation of STING. Within IDR, E^336^/E^337^ are the two essential residues for phase separation. (**C**) The domain structure of IRF3 and IRF7. IRF3 is composed of a DNA-binding domain (DBD), IDR region, IRF-associated domain (IAD) for dimerization, and signal response domain (SRD) containing serine residues 386 and 396 for activation. DBD, IDR, and IAD domains are required for its phase separation, while the SRD domain plays a negative role in its LLPS. IRF7 is composed of a DNA-binding domain (DBD), IDR region, IRF-associated domain (IAD) for dimerization, and signal response domain (SRD). DBD, IDR, and IAD domains are required for its phase separation, while the SRD domain plays a negative role in its LLPS. (**D**) The domain structure of the N protein of SARS-CoV-2. The N protein is composed of the N arm (N_IDR_), the N-terminal domain (NTD), the linker region (Linker_IDR_ or S/R rich motif), the C-terminal domain (CTD), and the C-terminal tail (C_IDR_). NTD and CTD bind with RNA, and CTD is the dimerization domain. N_IDR_ interacts with G3BP1, and Linker_IDR_ binds with NSP3. Within the Linker_IDR_ or S/R rich motif, the phosphorylation is important for the N protein to be involved in the viral transcription and replication. Linker_IDR_ and the CTD domains are essential for its phase separation. (**E**) The domain structure of the NSP8 protein of SARS-CoV-2. The NSP8 protein is composed of an N-terminal extension region (IDR: residues 1–76) and a C-terminal head region (interaction with NSP7 and NSP12). The N-terminal region is positively charged and responsible for binding with single-stranded nucleic acids and interaction with helicase NSP13. The C-terminal head region is crucial for binding with the NSP7 and NSP12 proteins to form the holo–RdRP complex. The N-terminal IDR region is required for its LLPS.

**Figure 3 pathogens-12-00243-f003:**
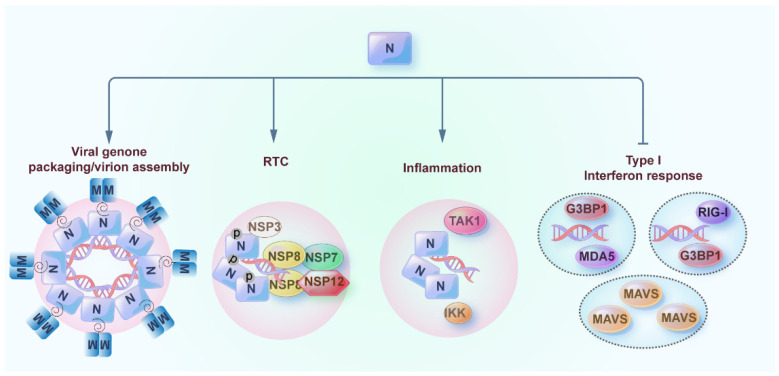
Functions of N protein phase separation. Phase separation of the N protein promotes viral genome packaging/virion assembly, RTC formation, and virus-induced inflammation. Phase separation of the N protein counteracts the RLR-MAVS mediated type I interferon response. Phase separation of N promotes viral genome packaging/virion assembly with the interaction between N and viral genomic RNA as well as the interaction between the N protein and M protein intravirion tail (left panel). RTC formation is driven by the N condensates with NSP3, NSP12, NSP7, and NSP8 (middle left panel). The phosphorylation of the N protein plays a critical switch between these two functions, with unphosphorylated N protein forming gel-like condensate to promote viral genome packaging and phosphorylated N protein facilitating viral replication and transcription. N protein condensates harboring TAK1 and IKK also facilitate inflammation (middle right panel). N protein condensates prevent the SG formation and aggregation of MAVS to inhibit the type I interferon response (right panel).

## Data Availability

Not applicable.

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
