# Peer review of "Phase Separation: The Robust Modulator of Innate Antiviral Signaling and SARS-CoV-2 Infection"

_pathogens, 2023, doi:10.3390/pathogens12020243_

Round 1

Reviewer 1 Report

In this review, Zheng and Gao summarize recent advances in understanding phase separation in virus-host interactions, focusing on phase separation in the regulation of antiviral innate immune responses and SARS-CoV infection. The review is comprehensive and sufficiently critical to be useful to researchers and collaborators in the field. The review is well written and needs only minor improvements to be useful to researchers who consider phase separation as a pathophysiological mechanism related to innate immunity and infection. A major shortcoming is the lack of graphical representation, which may be very helpful for those not necessarily concerned with the specific part of the field. Figures 1 and 2 are at the end of the main text and are not used to visualize the complex interactions described in the main text (sections 2-8). I suggest that these interactions be represented graphically and cross-referenced with Figure 1,

I would also suggest clarifying the abbreviation N used throughout the text (lines 205, 207, 212, 217, 219, 224, 225, 227, 255, 260, 277, 296, 301, 302, 309 ..). The abbreviation N is used in various contexts in the manuscript and may be confusing to the reader. I suggest using either "N protein" or perhaps "pN" to refer to the nucleocapsid protein of SARS-CoV-2.

Line 56, "compartalization" should be corrected.

Line 195 appears to be an incomplete sentence.

Reviewer 2 Report

The authors composed a concise review with clear figures about the role of phase separation in the innate immune response focusing on SARS-CoV-2 infection. There are only few things, which should be changed.

1.       All abbreviations such as PML-NB, G3BP1, USP15, PCBP2, PONDR, and NF2m should be introduced in the text and then used consistently (line 241: IDR has already been introduced in line 58-59).

2.       The authors should explain the biological function of membrane-less organelles especially Cajal bodies, PML-NB, and stress granules (line 55) as well as the structure and role of IDRs (line 58-59) in the introduction (line 55) with few sentences.

3.       The figures should be referenced and placed within and not behind the text.

Minors:

1.       Please replace “protein” by “proteins” in lines 14, 67, and 231.

2.       Please replace “condensate” by “condensates” in line 180.

3.       Please delete the “=” in line 217 (“RNA,=indicating”).

4.       Please replace “complex including” by “complex formation of” in lines 230 and 231.

5.       Please replace “under the” by “by a” in line 247.

6.       Please replace “aggregation of MAVS on the mitochondria. The aggregation of MAVS is liquid-like droplets.” by “aggregation of MAVS on the mitochondria in liquid-like droplets.” in lines 340 and 341 (figure legend of Figure 1).

7.       Please correct “Inteferon” in Figure 2.

Reviewer 3 Report

This is an interesting review on the relationships between viral and immune activities and phase separations. It adds to the great number of review on the SARS-CoV-2 and therefore it is important to be updated. However, my impression is that important studies published recently (e.g. last year), and relevant to the topic are missing. I specifically mention where the reviewing effort should be intensified, together with some missing or incorrect concepts and definitions that should be amended.

I suggest to include a figure that shows the domain architecture of some of the key proteins covered in this review (cGAS, STING, IRF3, IRF7, N and NSP8). As separate figure or by expanding figure 1. Placing no calls to figures throughout the text undervalue them.

References should be checked. For example, the Hirose et al. 2022 is missing. I assume that authors refer to Hirose, T., Ninomiya, K., Nakagawa, S. et al. A guide to membraneless organelles and their various roles in gene regulation. Nat Rev Mol Cell Biol (2022). https://doi.org/10.1038/s41580-022-00558-8. This is an excellent review on the LLPS phenomena and MLO. I understand the number of reviews on this topic are endless and citing all is impractical. However, just one reference seems a bit insufficient to me, thus therefore I would recommend to include additional ones for LLPS and its relationship with viral infections (e.g. PMID: 36464283, PMID: 3326071).

Line 113: “competition disrupts” instead of “competition destroys” 

Define cGAMP, PML-NB

I found several very recent reviews covering cGAS-STING and LLPS (PMID: 36579404, PMID: 35197578, PMID: 36410875, PMID: 34920941, PMID: 32424334). What is the distinct contribution of this one to the field, insufficiently covered by the previous? Why these previous reviews are not cited?

Line 195 “highly stable dimer” instead of “strong dimer”

Line 196. Is Zeng et al, 2020 the correct reference here?

Line 200. Dimerization? 

The bibliography of SARS CoV-2 nucleocapsid LLPS and particle packging is outdated. Several important studies on these topics have been published in 2022. This include structural cryoEM and cryoET data. The authors should revise significantly this section.

The section about NSP8 is weak and imprecise. For instance this protein a cofactor of RdPp, not the polymerase itself. Implications of NSP8 LLPS for viral activity are not discussed. Expand, integrate together with the previous section (making it more general to SARS-CoV-2 proteins experiencing PS), or remove.

The molecular details of G3BP1 and N protein interaction are known (PMID: 35240128) . It is surprising that this important information was missing in the review. As in the section 5, I think the literature is outdated, missing important references of 2022. 

Reviewer 4 Report

Generally, this manuscript was well written. I have several minor concerns as below:

1. Line 28-31 Please specify the individual types of viral nucleic acid for TRIF, MyD88, MAVS, and STING, respectively.

2. Line 54 "membrane-less" to "membraneless".

3. Line 62 "to promote" to "that promotes".

4. Line 64-66 Viral proteins also.......SARS-CoV-2 (Li et al., 2022)

The cited paper is for SARS-CoV-2 only, please also cite original references for measles virus and RSV in this regard.

5. Line 79 "the possible drugs" to "the potential drugs".

6. Line 146 Please define the full name for CARDs.

7. Line 337 "traffic" to "trafficking".

8. Figure 1: The phosphorylated TBK1 does not come from the cGAMP-TBK1-STING compartment?

Round 2

Reviewer 3 Report

Authors have made a significant effort to improve the quality of the manuscript and have addressed all my previous points correctly. Therefore, I recommend the publication of this review in its present form.